# A Bioengineering Approach for the Development of Fibroblast Growth Factor-7-Functionalized Sericin Biomaterial Applicable for the Cultivation of Keratinocytes

**DOI:** 10.3390/ijms23179953

**Published:** 2022-09-01

**Authors:** Ai Ai Lian, Yuka Yamaji, Kazuki Kajiwara, Keiko Takaki, Hajime Mori, Mervyn Wing On Liew, Eiji Kotani, Rina Maruta

**Affiliations:** 1Department of Applied Biology, Kyoto Institute of Technology, Sakyo-ku, Kyoto 606-8585, Japan; 2Biomedical Research Center, Kyoto Institute of Technology, Sakyo-ku, Kyoto 606-8585, Japan; 3Institute for Research in Molecular Medicine, Universiti Sains Malaysia, Minden 11800, Penang, Malaysia

**Keywords:** sericin, growth factors, protein stability, bioengineering, biomaterials, transgenic silkworms

## Abstract

Growth factors, including fibroblast growth factor-7 (FGF-7), are a group of proteins that stimulate various cellular processes and are often used with carriers to prevent the rapid loss of their activities. Sericin with great biocompatibility has been investigated as a proteinaceous carrier to enhance the stability of incorporated proteins. The difficulties in obtaining intact sericin from silkworm cocoons and the handling of growth factors with poor stability necessitate an efficient technique to incorporate the protein into a sericin-based biomaterial. Here, we report the generation of a transgenic silkworm line simultaneously expressing and incorporating FGF-7 into cocoon shells containing almost exclusively sericin. Growth-factor-functionalized sericin cocoon shells requiring simple lyophilization and pulverization processes were successfully used to induce the proliferation and migration of keratinocytes. Moreover, FGF-7 incorporated into sericin-cocoon powder exhibited remarkable stability, with more than 70% of bioactivity being retained after being stored as a suspension at 25 °C for 3 months. Transgenic sericin-cocoon powder was used to continuously supply biologically active FGF-7 to generate a three-dimensionally cultured keratinocyte model in vitro. The outcomes of this study propound a feasible approach to producing cytokine-functionalized sericin materials that are ready to use for cell cultivation.

## 1. Introduction

Keratinocyte growth factor, also known as fibroblast growth factor-7 (FGF-7), plays an important role in multiple cellular responses in epithelial cell lines [1,2]. The mitogenic activity of FGF-7 on keratinocytes has been well characterized [3,4], and some studies have reported its migratory effect on cells [5]. Following functional studies of FGF-7, this protein has been investigated for potential applications in regenerative medicine and tissue engineering [6,7,8]. Nonetheless, the inherently short half-life of growth factors in aqueous state has hindered their applications. Long-term availability of biologically active growth factors is required to prolong specific cellular activity that is important for a successful tissue-engineering application [9,10].

To improve the stability or bioavailability of growth factors, proteins have often been immobilized onto or incorporated into a variety of protein carriers, for example, encapsulation into the protein microcrystals derived from cypovirus [11,12], and immobilization onto polymeric biomaterials [13,14]. In recent decades, sericin, a group of adhesive proteins derived from silkworms, has emerged as a candidate for biopolymers owing to its non-mammal origin, biocompatibility, biodegradability, low immunogenicity [15,16], and availability of numerous polar amino-acid residues for modification [17,18]. A few studies have demonstrated the efficacy of using sericin to stabilize sericin-conjugated protein molecules [17,19]. Additionally, fibroblast growth factor-2 has been loaded onto regenerated sericin-based biomaterials used for the treatment of bone defects [10]. Nonetheless, sericin extracted from ordinary cocoons is usually fragmented and accompanied by the loss of some important biochemical properties [20,21]. Furthermore, the production, purification, and handling of labile proteins such as growth factors would constitute a significant portion of the final product cost.

The discovery of *piggyBac* transposon enables the generation of germline transgenic silkworms with silk gland cells expressing recombinant proteins that are then incorporated into cocoon shells [22]. Using the sericin-1 promoter encompassing a secretion-signal-peptide sequence (Ser1sp; Appendix A), recombinant proteins can be expressed along with the endogenous sericin proteins in middle silk gland (MSG) cells and secreted into MSG lumens as non-fusion proteins [23]. The biologically active proteins are then incorporated into presumed meshwork structure formed in the sericin layer of cocoons during silk spinning [24,25].

However, ordinary cocoons are composed of approximately 75% fibroin as the main silk proteins; thus, the capacity to increase the percentage of the recombinant protein in cocoons is limited. Additionally, fibroin is responsible for the fibrousness of cocoon silk fibers, which makes it difficult to pulverize through a mechanical process [26]. We previously established a transgenic silkworm line with impaired posterior silk glands (PSGs), in which fibroin protein synthesis was suppressed via the expression of cytotoxin pierisin-1 homolog derived from cabbage butterfly (pierisin-1A; P1A), which produced cocoons composed almost exclusively of sericin [27]. Despite having abnormal PSGs, this transgenic line can be propagated, and the sericin-cocoon-forming feature is heritable.

In the present study, we generated a transgenic silkworm line that lacks fibroin synthesis capabilities and expresses FGF-7 in MSG cells, and this was used as a biofactory for the generation of growth-factor-functionalized sericin materials. Next, we investigated the biological activities and storage stability of FGF-7 incorporated into sericin-cocoon powder. The application efficiency of sericin-cocoon powder containing growth factors was investigated by generating a three-dimensional (3D) epidermal model in vitro, which required a sustained supply of biologically active FGF-7 for 2 weeks.

## 2. Results

### 2.1. Establishment of Transgenic Silkworm Line Spinning Sericin Cocoons Incorporating H1/FGF-7

To obtain silkworm cocoons containing FGF-7 in the sericin layer, we generated transgenic silkworm line S1sp-F7 carrying FGF-7 cDNA-fused with the polyhedra-encapsulation signal helix-1 sequence (H1/FGF-7) and regulated by the sericin-1 promoter (Figure 1A). The fusion with the H1 peptide facilitated the encapsulation of foreign proteins using microcrystals of cypovirus polyhedrin (also known as polyhedra) and showed no adverse effects on the biological activities of the fused proteins, including FGF-7 [11,28]. The stabilization effect of cypovirus polyhedra on encapsulated proteins against the hostile environment [11,12,28] drove us to investigate the stability of H1/FGF-7 as a combined effect of encapsulation by polyhedra and incorporation into sericin cocoons. Therefore, transgenic silkworm line S1sp-F7/poly producing cocoons with sericin layers incorporated with co-expressed H1/FGF-7 and polyhedrin was generated as described in the Appendix A.

The transgene inserted into the S1sp-F7 silkworm genome was found to be flanked by sequences from chromosome 1 (Figure 1B), as determined with inverse PCR analysis. Transgenic line S1sp-F7 was then mated with the previously established sericin cocoon (SC)-forming line *w1-pnd^P1A269/P1A269^* (renamed FH-P1A to convey information on the promoter (fibroin heavy-chain promoter; FH) and expressed protein (P1A)) [27] to generate the S1sp-F7 × FH-P1A line, which produced sericin cocoons incorporating H1/FGF-7 (F7-SC; Figure 1A).

To obtain sericin cocoons incorporating polyhedra-encapsulated H1/FGF-7 (F7/poly-SC; Appendix A), transgenic line S1sp-F7/poly was mated with the FH-P1A line. However, the crystallization of polyhedrin was not attained in the sericin cocoon shells of the mated line S1sp-F7/poly × FH-P1A or sericin layers of the ordinary cocoon shells obtained from the S1sp-F7/poly line (Appendix A). Hence, the results of the functional study on F7/poly-SC are only provided in the Appendix A.

The presence of H1/FGF-7 in F7-SC was investigated via immunoblotting using an anti-FGF-7 antibody (Figure 2). Polyhedra crystals encapsulating H1/FGF-7 (F7-PH crystals; Figure 2, Lane 2) were used as a positive control for Western blot analyses; a band with an approximate molecular mass of 30 kDa, which was consistent with the size of FGF-7 fused with the H1 peptide [11], was detected, and a smaller band observed around 20 kDa suggested the degradation of H1/FGF-7. Sericin cocoons collected from the FH-P1A line (SC; Figure 2, Lane 3) and polyhedra obtained from the Sf21 cell culture infected with recombinant baculovirus carrying polyhedrin cDNA [11] (CPH; Figure 2, Lane 1) were used as negative controls, and no protein bands were detected. The incorporation of H1/FGF-7 into the sericin cocoon shells obtained from the S1sp-F7 × FH-P1A line (F7-SC; Figure 2, Lane 4) was confirmed by the presence of a band around 30 kDa. As observed in the positive-control F7-PH crystals (Figure 2, Lane 2), a degraded H1/FGF-7 band was also detected in F7-SC (Figure 2, Lane 4). The slight difference in the H1/FGF-7 molecular mass observed between F7-PH crystals and F7-SC (Figure 2, Lane 2 and Lane 4) might have been caused by the varied sample compositions resulting from the different treatments of polyhedra and sericin samples (see Materials and Methods, Section 4.5). In addition, the probable cleavage site of the sericin-1 promoter-derived secretion signal peptide (consisting of the first 21 N-terminal amino-acid residues) at positions 19 and 20 (Appendix A) [29] also likely affected the H1/FGF-7 protein size detected in F7-SC, as the two amino-acid residues remained. The band intensity ratio of H1/FGF-7 detected with 5 × 10^4^ F7-PH and 10 µg of F7-SC was 1:1.3, as seen using ImageJ analysis. This result implied that the amounts of F7-PH and F7-SC used in the Western blot analyses contained comparable levels of H1/FGF-7. In order to use an approximate amount of H1/FGF-7 from F7-PH and F7-SC for the following biological activity assays, an amount ratio of F7-PH and F7-SC similar to that applied in the current Western blot analyses was used.

### 2.2. Release Profile of H1/FGF-7 from F7-SC Powders in Aqueous Media

Understanding the release profile of incorporated proteins from the materials is important for the development of a successful protein carrier. Hence, we investigated the release of H1/FGF-7 from lyophilized F7-SC powder into a defined medium formulated for the cultivation of keratinocytes, DK-SFM (defined keratinocyte serum-free medium) and keratinocyte-conditioned DK-SFM (KCM). Sericin-cocoon powders were added to the cell-culture inserts that were placed into the culture plate. The plate was than incubated at 37 °C without agitation, and the amount of H1/FGF-7 released was measured using the ELISA every 3 or 4 days up to 21 days for KCM and up to 14 days for DK-SFM (Figure 3).

The total amounts of H1/FGF-7 released from 1 mg of F7-SC powder into KCM and DK-SFM were 29.5 and 19.8 ng, respectively (Figure 3). The accumulated amount of H1/FGF-7 released from F7-SC powder into KCM over the first three days was 12.2 ng, which accounted for 41.3% of the total amount released. Over the next 4 days, 9.7 ng was detected, followed by 4.5 ng and 2.3 ng, which were detected from day 7 to day 10 and from day 10 to day 14, respectively. The remaining 0.8 ng of H1/FGF-7 was released during the last week. The release pattern of H1/FGF-7 into DK-SFM was similar to that observed in KCM but with a lower amount of H1/FGF-7 (Figure 3).

### 2.3. Biological Activities of H1/FGF-7 Incorporated into Sericin-Cocoon Powder

Growth factor FGF-7 is involved in various biological activities of epithelial cells, such as proliferation and migration [30,31]. Thus, we investigated the proliferation and migration of normal human epidermal keratinocytes (NHEKs) in the cultures containing H1/FGF-7 released from F7-SC powder added into cell-culture inserts. The proliferation rates of NHEK cells cultured in DK-SFM supplemented with F7-SC powder, F7-PH crystals, commercial recombinant human FGF-7 (rhFGF-7), and SC powder were measured as the relative viable cell numbers on day 3 (Figure 4). Dulbecco’s phosphate-buffered saline (−) (PBS) was added to the culture of the non-treated control. As shown in the release profile of F7-SC powder (Figure 3), the accumulated releases of H1/FGF-7 from 50 and 200 µg of F7-SC powder were 1.5 and 6.0 ng, which were comparable to the amount range of the positive controls used (1 to 10 ng of rhFGF-7). Hence, the amount of F7-SC powder in this range was chosen for the subsequent biological activity assays.

The cultures of keratinocytes supplemented with 200 µg of SC powder did not show significant differences in the proliferation rate compared to the non-treated control (*p* > 0.05). Conversely, an increase in the viable cell number was measured in NHEK cultures supplemented with rhFGF-7 at all examined doses (1, 5, and 10 ng) or 5 × 10^5^ cubes of F7-PH crystals. The addition of 1 ng of rhFGF-7 resulted in an approximately 0.7-fold increase in cell proliferation in comparison with the non-treated control (*p* < 0.001). Cell proliferation was slightly enhanced, 0.3-fold, when the amount of rhFGF-7 was increased from 1 to 5 ng (*p* > 0.05) and was further enhanced, 2.3-fold, when rhFGF-7 was increased from 5 to 10 ng (*p* < 0.001). The dose-dependent effect was also observed in NHEK cultures supplemented with F7-SC powder. The addition of 50 µg of F7-SC powder to NHEK cultures resulted in approximately 2.3-fold cell proliferation compared with the non-treated control (*p* < 0.001). Cell proliferation increased from 2.3-fold to 2.9-fold (*p* < 0.01) when the amount of F7-SC powder was increased from 50 to 100 µg and further improved, 0.5-fold, when 200 µg was used (*p* < 0.001).

The migration of keratinocytes in cultures treated with F7-SC powder was examined using an in vitro scratch assay, and the wound closure percentage after 24 h of treatment was determined (Figure 5). The non-treated control showed a wound closure percentage of 26.7%, and the addition of 100 µg of SC powder marginally improved wound closure to 33.6%. The addition of 5 × 10^5^ cubes of F7-PH crystals and 10 ng of rhFGF-7 resulted in wound closures of 50.5% and 46.8%, respectively. The enhancement of wound closure induced by F7-PH crystals and rhFGF-7 was statistically insignificant compared to that observed in the cultures of the non-treated control (*p* > 0.05). The cultures treated with 100 µg of F7-SC powder 2.0-fold improved the wound closure percentage when compared with the non-treated control (*p* < 0.05). Taken together, the wound-closure results suggested that FGF-7 supplemented in different forms (F7-SC powder, F7-PH crystals, and rhFGF-7) was capable of stimulating the migration of NHEK cells, although the effect was weak. This result was consistent with that reported in other studies. FGF-7 has a migratory effect on keratinocytes; however, the effect is relatively weak and, sometimes, insignificant [32,33].

### 2.4. Storage Stability of H1/FGF-7 Incorporated into Sericin-Cocoon Powder

Non-modified FGF-7 is highly unstable when stored at elevated temperatures, especially in aqueous form [34]. To investigate the storage stability of H1/FGF-7 incorporated into the presumed native sericin meshwork structure formed during cocoon spinning, we measured the proliferation rate of NHEK cells induced by an F7-SC suspension that had been stored at 25 °C for 1 week and compared it to that induced by corresponding samples stored at −20 °C (Figure 6A). Several studies have shown the efficiency of cypovirus polyhedra in protecting encapsulated proteins from stress factors that can cause protein denaturation, such as elevated temperature [12,35]. Therefore, H1/FGF-7 encapsulated in polyhedra (F7-PH crystals) was included in this study for comparison. Conversely, the non-modified rhFGF-7 solution was used as a control.

The proliferative activity observed with non-modified rhFGF-7 incubated at 25 °C for 1 week was 69% lower than that of its counterpart at −20 °C (*p* < 0.001). The change in proliferative activity was −11% for F7-SC powder and +6% for F7-PH crystals when comparing the samples stored for 1 week at 25 °C with the corresponding samples at −20 °C. Nonetheless, neither the changes in the proliferative activity of F7-SC powder nor those in F7-PH crystals after incubation at elevated temperatures were statistically significant (*p* > 0.05). The marginal increase in proliferative activity observed with F7-PH crystals kept at 25 °C, as compared with its counterpart at −20 °C, was likely caused by the elevated temperature. An increase in temperature may have accelerated protein release from polyhedra and subsequently improved the proliferation of NHEK cells. The marginal and statistically insignificant decrease in the proliferative activity of F7-SC powder stored at 25 °C suggests that the presumed naturally formed sericin meshwork structure was effective in protecting the biological activity of incorporated H1/FGF-7. In addition, we investigated the long-term storage stability of F7-SC powder for up to 3 months (Figure 6B). The proliferative activity of F7-SC suspension and free rhFGF-7, which were stored at 25 °C for 3 months prior to the assays, showed reductions of 26% (*p* < 0.01) and 79% (*p* < 0.001), respectively, when compared with the corresponding samples stored at −20 °C (Figure 6B).

### 2.5. Three-Dimensional Cultivation of Keratinocytes In Vitro

The generation of a 3D epidermal model in vitro requires a sustained supply of biologically active growth factors to induce the proliferation and differentiation of keratinocytes. To investigate the efficacy of using H1/FGF-7-functionalized sericin-cocoon powder as a long-term growth-factor-releasing material, keratinocytes were 3D-cultured on collagen gel containing F7-SC powder. Keratinocytes cultured on non-treated collagen gel were supplemented with 300 ng of rhFGF-7 and used as a control. To avoid the deprivation of active rhFGF-7 in the control culture, fresh growth factors were supplemented twice per week until the end of cultivation.

Hematoxylin and eosin (HE) staining of the 3D-cultured keratinocytes treated with rhFGF-7 showed that upper layers composed of enucleated cells were relatively thin and presented irregular thickness (Figure 7A, right panel), indicating that a poor stratum corneum layer had formed. The failure of rhFGF-7-treated and 3D-cultured keratinocytes to form proper stratum corneum layers appears to be a result of insufficient rhFGF-7 for the support of the proliferation and differentiation of keratinocytes. On the other hand, cells cultivated on collagen gel containing F7-SC powder were stratified, with obvious upper layers containing enucleated cells and lower layers with observed nuclei (Figure 7A, left panel). The formation of the enucleated cell layers was a result of NHEK cell differentiation, thus implying that a sufficient amount of active H1/FGF-7 was released from F7-SC powder into the collagen gels to support the proliferation and differentiation of the cells.

Next, we performed immunofluorescent staining to examine the presence of differentiation markers in keratinocytes 3D-cultured on collagen gel containing F7-SC powder (Figure 7B). Loricrin and filaggrin in the stratum corneum and stratum granulosum layers, together with keratin 10 in the stratum spinosum layer and keratin 14 in the stratum basale layer, were detected through incubation with their respective antibodies (Figure 7B; loricrin and keratin 14 on the left panel and filaggrin and keratin 10 on the right panel). Loricrin and filaggrin were detected in the upper layers (above the dashed lines), where nucleated cells were not observed. Keratin 10 and keratin 14 were observed in the lower layers of the 3D-cultured keratinocyte model. The presence of nucleated cells in the lower layers of the models was evidenced with the observed 4′,6-diamidino-2-phenylindole (DAPI)-stained nuclei. The immunofluorescent staining results showed that the 3D-cultured keratinocytes expressed the anticipated differentiation markers.

The results showed that the keratinocytes 3D-cultured on collagen gels containing F7-SC were differentiated to form obvious stratum corneum layers. However, further studies to optimize the cell seeding density and medium composition would be required for the generation of an epidermal model with distinguishable nucleated cell layers which comprise stratum granulosum, stratum spinosum, and stratum basale [36].

## 3. Discussion

In the present study, we generated a transgenic silkworm line that produced sericin cocoons functionalized with H1/FGF-7 (F7-SC) by mating the transgenic Ser1-F7 and FH-P1A lines (Ser1-F7 × FH-P1A line; Figure 1A and Figure 2). The silkworm FH-P1A line that lacks fibroin synthesis function was generated by expressing butterfly cytotoxin P1A in PSG cells [27]. The potential of sericin material as the FGF-7 carrier in the 2D and 3D keratinocyte cultivation systems was evaluated using H1/FGF-7-incorporating sericin-cocoon powder.

We showed that the release of H1/FGF-7 from 3% (*w*/*v*) F7-SC powder into KCM was generally higher and longer than that into DK-SFM (Figure 3). The presence of alkaline proteases and neutral proteases is known to facilitate the degradation of sericin [37]. The proteomic analysis of the KCM revealed that a myriad of proteins from different classes, such as metabolic enzymes, proteasomes, and matrix metalloproteinases, were released by keratinocytes into the medium [38]. These enzymes were likely involved in the degradation process of sericin, resulting in a higher amount of H1/FGF-7 being released into KCM than that into DK-SFM (Figure 3). In addition, recombinant proteins expressed by the MSG systems were suggested to be tightly trapped in the presumed meshwork structure formed in the sericin layer but could be easily released into the aqueous buffer preceding sericin degradation even at the low temperature of 4 °C [23]. Thus, the release of H1/FGF-7 observed in DK-SFM could be also attributed to the dissolution of H1/FGF-7 into water, which was absorbed into the presumed sericin meshwork structure. Such evidence suggests that growth-factor-functionalized sericin-cocoon powder with no treatment of H1/FGF-7 extraction could be introduced into the cultivation condition for keratinocytes described in this study.

We found that the H1/FGF-7 released from F7-SC powder was capable of enhancing the proliferation and migration of keratinocytes (Figure 4 and Figure 5). Additionally, soluble silk sericin has also been shown to facilitate the migration of epithelial cells via the activation of the c-Jun pathway [39]. Thus, the dissolved sericin proteins from F7-SC powder could have passed through the filters of the cell-culture inserts to act synergistically with H1/FGF-7 to enhance wound closure. The migration of keratinocytes is a pivotal event during re-epithelization in the wound-healing process [5,6,40]. This finding suggests that preparing wound-dressing materials from sericin cocoons functionalized with H1/FGF-7 could benefit from the biological activities of the growth factor and the cell migration induced by sericin.

The recombinant H1/FGF-7 expressed by MSG cells and incorporated into sericin-cocoon powder was shown to have a good retention of cell proliferative activity after being stored at an elevated temperature (25 °C) and in an aqueous state (Figure 6). The stabilization effects of sericin on clinically important proteins against protease digestion and thermally induced degradation have been reported in several bioconjugation studies [17,19,41]. Sericin-conjugated anti-leukemic enzyme L-asparaginase has a half-life of more than a year in aqueous form at 4 °C, which is greatly improved compared with the 1-week half-life of native L-asparaginase [17]. Despite the elevated half-life, activity loss due to the bioconjugation process was inevitable, and a series of purification processes were required following bioconjugation. The protein stabilization effect observed in the sericin-derived bioconjugates was attributed to the steric hindrance effects [17,19] resulting from the chemical modification of the amino-acid residues of the protein. The H1/FGF-7 protein was not chemically modified by sericin but physically interacted with sericin through incorporation into the presumed meshwork structure of sericin. Therefore, another explanation would be needed for the mechanism of H1/FGF-7 stabilization in sericin cocoons. A recent molecular-dynamics simulation study investigating the thermodynamic stability of coiled-coil protein GCN4-P1 challenges the classical concept that water-mediated hydrophobic interactions are factors leading to protein folding; instead, these interactions could destabilize folded proteins [42]. Thus, we speculate that the limited access to water molecules due to the shielding effect from the sericin meshwork structure is one of the factors contributing to the stabilization of incorporated H1/FGF-7. Additionally, carbohydrate-derived polymers and anionic inorganic polymers such as heparin and polyphosphates have been reported to be effective in reducing the aggregation of FGF-7 in water [34], which would consequently result in the loss of bioactivity. Therefore, it appears that the dissolved intact sericin acted as a biopolymer to protect released H1/FGF-7 by reducing or minimizing protein aggregation. Such evidence suggests that the sericin cocoon can stabilize the proteins incorporated into it.

The current study revealed a 3D-cultured keratinocyte model with the stratification and expression of anticipated differentiation markers generated by 3D-cultivating keratinocytes on collagen gel containing F7-SC powder (Figure 7A, left panel). The use of animal models for toxicological and immunological testing was once the mainstream. However, the ethical issues and histological differences of animal skins from human skins have caused an increasing need for alternative testing systems [43]. Epidermal or skin models have been proposed as promising alternatives for the testing of toxicological [44,45], allergenicity [46], and inflammatory [47] responses triggered by test compounds. Considering the functions of 3D epidermal models in compound testing, materials used for the generation of models need to be biocompatible to avoid complications of the test results. For a long time, sericin has been blamed for the immunological responses associated with silk-derived products. Meanwhile, a growing number of studies have proved the biocompatibility of sericin in terms of inflammatory responses, allergenicity, and immunogenicity through a series of experiments. For example, the densities of macrophages and neutrophiles infiltrated into sericin hydrogels implanted in mice were similar to those of the control groups; the levels of IgE and IgG produced by mice injected with sericin were not increased [15], and macrophages cultured with soluble sericin did not show significant increases in TNF production as compared with the controls [16]. In addition, silkworm-derived allergens, such as arginine kinase [48], which could have contaminated the silk products during the filature and degumming processes, were found to be the possible causes of immunological responses. Cocoon-derived fibroin have shown mechanical properties that are attractive to material science [49]. However, a previous study raised concerns regarding the immunological responses triggered by insoluble fibroin particles from α-chymotrypsin-digested fibroin fibers [16]. As fibroin-free sericin materials can be prepared from sericin cocoon shells without filature and degumming steps, obtaining allergen-free and immunological-inert sericin materials is possible. Therefore, sericin cocoons functionalized with H1/FGF-7 have good potential to be used as growth-factor-releasing materials for the generation of 3D epidermal models that are appropriate for compound testing. The above evidence collectively demonstrates that the proposed bioengineering approach is feasible for producing growth-factor-functionalized sericin materials that can be used to prepare small-sized particles with potential application in various cell cultivation systems.

Additionally, the possibility to prepare allergy-free sericin powders from fibroin-free cocoon shells is beneficial for in vivo applications. Therefore, growth-factor-functionalized sericin materials have the potential to be used as protein carriers for tissue engineering, which require long-term availability of specific growth factors to induce the regrowth of diseased or injured tissues.

## 4. Materials and Methods

### 4.1. Silkworm Strains

The *w1-pnd* strain (non-pigmented and non-diapausing) was used to generate transgenic silkworms. Silkworm larvae were reared on an artificial diet (Kimono Brain Co., Ltd., Niigata, Japan) at 25 °C.

### 4.2. Cultivation of NHEK Cells

NHEK cells were cultivated in Humedia KG-2 medium supplemented with proliferation additives according to the manufacturer’s instructions (Kurabo Industries Ltd., Osaka, Japan) and incubated at 37 °C and 5% CO_2_.

### 4.3. Generation of Transgenic Silkworm Lines S1sp-F7 and S1sp-F7 × FH-P1A

Transgenic silkworm line S1sp-F7 was created to produce cocoons shells incorporating H1/FGF-7 in the sericin layer of silkworm cocoons (Figure 1A). Microinjection was performed as previously described [12,28,50,51] to generate S1sp-F7. In brief, the constructed vector, pBacMCS (hr3-Ser1sp-H1/FGF-7; 3 × P3-EGFP-SV40) (Appendix A), was co-injected with a helper plasmid that encodes the *piggyBac* transposase into pre-blastodermic *w1-pnd* embryos [52]. Silkworm adults obtained from the microinjected eggs (G0) were sib-mated, and the resulting eggs (G1) were screened for the expression of vector-derived marker gene EGFP, which is driven by 3 × P3 at the late embryonic stage. Transgenesis was further confirmed with inverse PCR, as described in the Appendix A (Section 2.4) using primers No.17 to 20 listed in Appendix A.

The established S1sp-F7 line was further mated with FH-P1A [27] to generate the S1sp-F7 × FH-P1A line producing sericin cocoons containing H1/FGF-7 (Figure 1A). Homogenous S1sp-F7 × FH-P1A silkworms, as indicated by the manifestation of strong marker traits, were isolated and propagated. The presence of the H1/FGF-7 transgene was further confirmed with a qPCR analysis of shredded skin from larvae. The transgenic line was maintained for more than 2 years in the laboratory through sib-mating and constant screening.

### 4.4. Preparation of Sericin-Cocoon Powders

The sericin cocoons collected from transgenic lines FH-P1A and S1sp-F7 × FH-P1A were pulverized for 10 s using a blender (Wonder Blender Osaka Chemical Co., Ltd., Osaka, Japan), lyophilized, and kept at −20 °C until further use.

### 4.5. Western Blot Analysis of H1/FGF-7

The presence of H1/FGF-7 in transgenic silkworm cocoons was examined through Western blotting. SC and F7-SC cocoon powders were dissolved in 8 M lithium bromide (LiBr) at 0.05 g/mL. SC and F7-SC (10 µg of each) were separated on 12.5% e-PAGEL (ATTO, Co., Ltd., Tokyo, Japan) and transferred onto a PVDF membrane (GE Healthcare Bioscience). Following blocking with 8 mL off Blocking One (Nacalai Tesque Inc., Kyoto, Japan), the membrane was incubated with a rabbit-derived anti-rhFGF-7 antibody (ReliaTech GmbH, Wolfenbüttel, Germany) diluted using PBS at a ratio of 1:2500. After overnight incubation with the primary antibody, the membrane was washed, blocked, and incubated with a (1:2500-diluted) secondary antibody (goat anti-rabbit IgG conjugated with horseradish peroxidase; Bio-Rad Laboratories, Hercules, CA, USA). The protein band of H1/FGF-7 was visualized using Peroxidase Stain DAB Kit (Brown Kit, Nacalai Tesque). In total, 50,000 cubes of CPH were used as a negative control, and 5 × 10^4^ cubes of H1/FGF-7-encapsulated polyhedra crystals (F7-PH crystals) were used as a positive control. The preparation and processing of polyhedra were described in a previous report [11]. Band intensity was analyzed using ImageJ software (LOCI, University of Wisconsin, Madison, WI, USA).

### 4.6. Release Profile of H1/FGF-7 from F7-SC Powder

The release of H1/FGF-7 from F7-SC powders into KCM and DK-SFM was examined. KCM was prepared by cultivating NHEK cells in DK-SFM at a cell density of 3 × 10^4^ cells/mL. After 48 h of cultivation, the medium was collected and filter-sterilized. KCM or DK-SFM (500 µL per well) was pipetted into a 24-well plate (Greiner Bio-One GmbH, Kremsmünster, Austria), and F7-SC powder was added to the 0.4 µm cell-culture inserts (Greiner Bio-One GmbH) placed on the wells at a final concentration of 0.04% (*w*/*v*). The media were changed on days 3, 7, 10, 14, 17, and 21. The concentration of H1/FGF-7 in the collected media was determined using enzyme-linked immunosorbent assays (ELISA; Quantikine Human KGF/FGF-7; R&D Systems, Minneapolis, MN, USA). The ELISAs were performed according to the manufacturer’s instructions. KCM and DK-SFM were used as blanks for the corresponding assays.

### 4.7. Effect of H1/FGF-7 on Proliferation of NHEK Cells

NHEK cells were plated onto 24-well plates at a density of 7500 cells per well. Suspensions of SC and F7-SC powders in 50 µL of PBS were added at the final amounts of 50, 100, and 200 µg per well. Suspensions of F7-PH crystals in 50 µL of PBS were added at 5 × 10^5^ cubes per well. Cell-culture inserts were used to hold the cocoon and polyhedra samples. Different amounts (1, 5, and 10 ng) of rhFGF-7 (FUJIFILM Wako Pure Chemical Corporation, Osaka, Japan) in 50 µL of PBS were added directly to the positive-control wells. Culture supplemented with 50 µL of PBS was used as a control. Cells were cultured in 500 µL of DK-SFM medium for 3 days at 37 °C and 5% CO_2_. On day 3, the inserts were discarded, and the spent medium was replaced with DK-SFM containing 10% WST-8 solution (Cell Counting Kit-8; Dojindo Molecular Technologies, Inc., Rockville, MD, USA) and further incubated for 4 h. The medium (100 µL) was then transferred to a 96-well plate, and the absorbance was measured at 450 nm.

### 4.8. Effect of H1/FGF-7 on Migration of NHEK Cells

NHEK cells at a density of 7500 cells per well were seeded onto 24-well plates and cultured in Humedia KG-2 at 37 °C and 5% CO_2_ until confluent. The cells were then cultivated in a starvation medium (DK-SFM) for 16 h. Thereafter, the culture was scratched to make a cross at the center of the wells, and the floating cells were removed via medium replacement. A total of 100 µg of SC powder, 100 µg of F7-SC powder, or 5 × 10^5^ cubes of F7-PH crystals in 50 µL of PBS was then added into the cell-culture inserts in the designated wells, and 10 ng of rhFGF-7 was added directly into the medium of the positive-control wells. Culture supplemented with 50 µL of PBS was used as a control. The plates were further incubated for 24 h. The width of the scratched wound was observed using an inverted microscope (Olympus IX73; Olympus, Tokyo, Japan) 0 and 24 h after incubation. The percentage of wound closure after 24 h of treatment was determined using ImageJ software.

### 4.9. Storage Stability of H1/FGF-7 Incorporated into Sericin-Cocoon Powder

Suspensions of F7-SC powder and F7-PH crystals in PBS were prepared at concentrations of 2 mg/mL and 1 × 10^7^ cubes/mL, respectively. Commercial rhFGF-7 powder containing Na_3_PO_4_ and NaCl was reconstituted in water to obtain a final protein concentration of 50 µg/mL. All samples were thoroughly mixed and equally divided into two groups. One set of the samples was stored at −20 °C, and the second set was stored at 25 °C. After 1 week of storage, the samples were used for the NHEK cell proliferation assay, as described in Section 4.7. F7-SC powder and F7-PH crystals were used at 100 µg and 5 × 10^5^ cubes per well, respectively. Control rhFGF-7 was first diluted to 0.2 ng/µL using PBS and added at 10 ng per well. The biological activity of the samples was determined as the viable cell number measured in the supplemented cultures. The activity of the samples stored at −20 °C was set to 100%, and the relative activity of the corresponding samples stored at 25 °C was determined.

### 4.10. Three-Dimensional Cultivation of NHEK Cells

A total of 800 µg of F7-SC powder was suspended in 300 µL of collagen solution (Cellmatrix Type I-A; Nitta Gelatin Inc., Osaka, Japan) and added into cloning rings in a 6-well plate. An equal volume of collagen solution only was used as a control. The plate was incubated at 37 °C for 1 h to induce gelation. DK-SFM (3 mL) was added around the rings, and 2 × 10^5^ NHEK cells were deposited on the collagen gels. For control wells, 300 ng of rhFGF-7 was added into the medium. The culture was incubated at 37 °C and 5% CO_2_. After two days of incubation, the rings were removed, and the cells were washed with PBS. Differentiation medium (1 mL of DK-SFM containing 1.2 mM Ca^2+^) was carefully added to the surroundings of the collagen gels. The top of the culture was left uncovered by the medium to allow the cells to grow at the air–liquid interface. The basal medium, DK-SFM, was changed twice per week, and fresh rhFGF-7 was supplemented to the positive-control well at the same frequency. On day 14, the medium was removed, and the cells were fixed with 4% paraformaldehyde overnight at 4 °C to create a 3D-cultured keratinocyte model. The model was washed with PBS before being embedded in the Optimal Cutting Temperature compound (Sakura Finetek Japan Co., Ltd., Tokyo, Japan). The model was then cryo-sectioned into 20 µm thick slices and fixed onto glass slides for HE staining and immunofluorescent staining. The cryosections were stained with hematoxylin and intensively washed with water before eosin staining. Subsequently, the glass slides were dehydrated using progressively higher percentages of ethanol. Finally, the slides were cleaned with Clear Plus (Pharma Co., Ltd., Aomori, Japan) and fixed with a mounting solution (PARAmount; Pharma).

For immunofluorescent staining, the slides were incubated with a blocking solution (5% normal goat serum; Vector Laboratories, Inc., Newark, CA, USA) and incubated with primary antibodies, rabbit-derived loricrin (BioLegend, San Diego, CA, USA) and keratin 10 (Abcam, Cambridge, UK), and mouse-derived keratin 14 (Invitrogen) and filaggrin (Santa Cruz Biotechnology Inc., Richmond, TX, USA). After washing with PBS, the sections were incubated with a mixture of Alexa Fluor 488-conjugated anti-rabbit IgG (Thermo Fischer Scientific, Waltham, MA, USA) and Alexa Fluor 594-conjugated anti-mouse IgG (Thermo Fischer Scientific). Nucleic acid was stained with DAPI. The immunofluorescent imaging of the cryosections was conducted using a confocal laser scanning microscope (FV1000-IX81; Olympus) with a 40× objective lens.

### 4.11. Statistical Analysis

Statistical analyses were performed using SigmaPlot 12.0 (Systat Software, Richmond, CA, USA). Data are presented as means ± standard deviations (SDs) of three independent experiments. A one-way analysis of variance followed by Tukey’s post hoc test for pairwise comparisons was conducted for cell proliferation and migration assays. Student’s t-test was performed for the storage-stability assay. Differences with *p* < 0.001, 0.01, or 0.05 were regarded as significant.

## 5. Conclusions

This study reports the generation of germline transgenic silkworms (*Bombyx mori*) producing sericin cocoons incorporating expressed FGF-7. The incorporated growth factor was biologically active and exhibited long-term stability in the aqueous state and at an elevated temperature. A 3D-cultured keratinocyte model was successfully generated in vitro using FGF-7-functionalized sericin-cocoon powder to continuously supply biologically active FGF-7 during cultivation. This study proposes a bioengineering approach to simultaneously express and incorporate biologically active proteins into sericin-cocoon powders that play the role of carriers of the proteins.

## Figures and Tables

**Figure 1 ijms-23-09953-f001:**
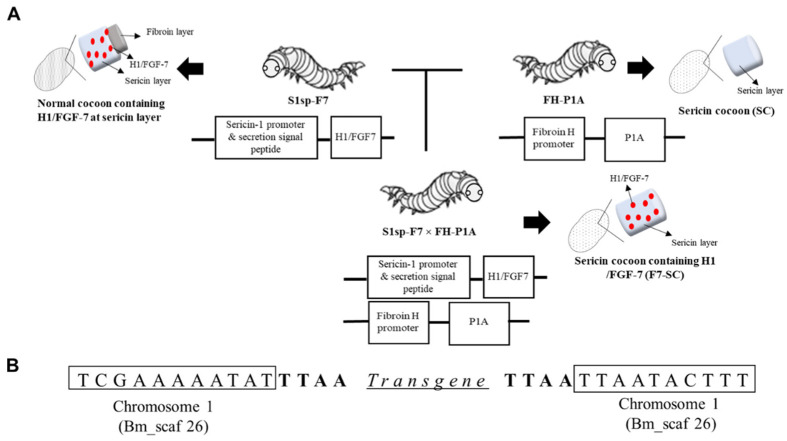
Generation of transgenic silkworm lines. (**A**) Transgenic silkworm lines and the corresponding cocoons. Transgenic line S1sp-F7 producing cocoons containing H1/FGF-7 in the sericin layer was mated with the FH-P1A line producing cocoons solely containing sericin (SC) to generate the S1sp-F7 × FH-P1A line. The resultant line, S1sp-F7 × FH-P1A, produced sericin cocoons incorporating H1/FGF-7 (F7-SC). (**B**) Sequences from chromosome 1 flanking the inserted transgene in line S1sp-F7.

**Figure 2 ijms-23-09953-f002:**
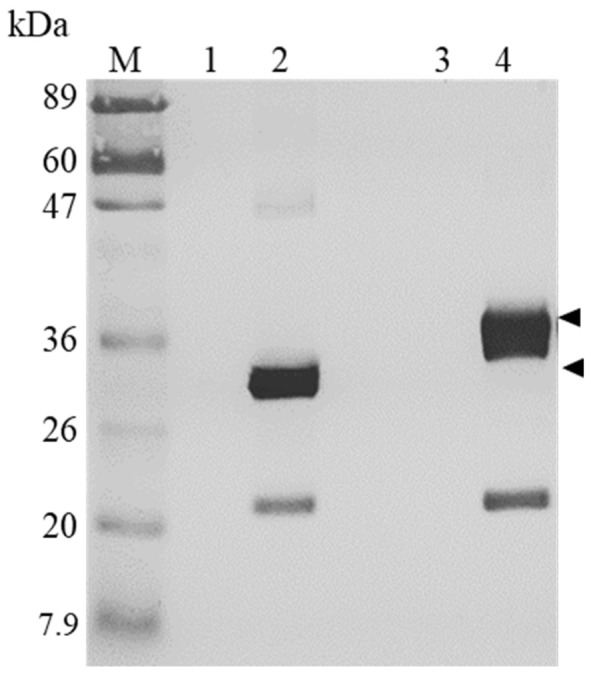
Western blot analyses of H1/FGF-7. (Lane M) Protein molecular weight marker; polyhedra (5 × 10^4^) obtained from Sf21 cell culture infected with (Lane 1; CPH) recombinant baculovirus carrying cDNA of polyhedrin and (Lane 2; F7-PH) recombinant baculovirus carrying cDNA of H1/FGF-7 and polyhedrin. Sericin cocoons of 10 µg derived from (Lane 3; SC) the FH-P1A line and (Lane 4; F7-SC) the S1sp-F7 × FH-P1A line. Arrow heads indicate the position of H1/FGF-7.

**Figure 3 ijms-23-09953-f003:**
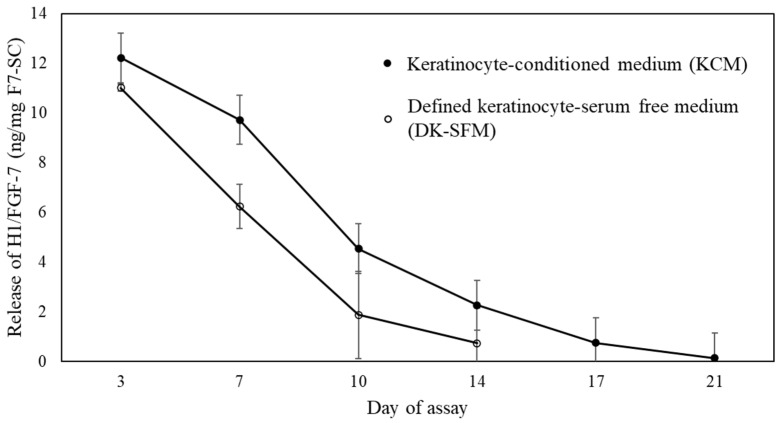
Release profile of H1/FGF-7 from F7-SC powder suspended in aqueous media. Powder of F7-SC was added into keratinocyte-conditioned medium (KCM; •) and DK-SFM (◦) at 0.04% (*w*/*v*) and incubated at 37 °C. Media were replaced at the indicated time points, and the amount of H1/FGF-7 released into the collected media was measured using the ELISA. Data are shown as means ± standard deviations (SDs) of triplicate assays.

**Figure 4 ijms-23-09953-f004:**
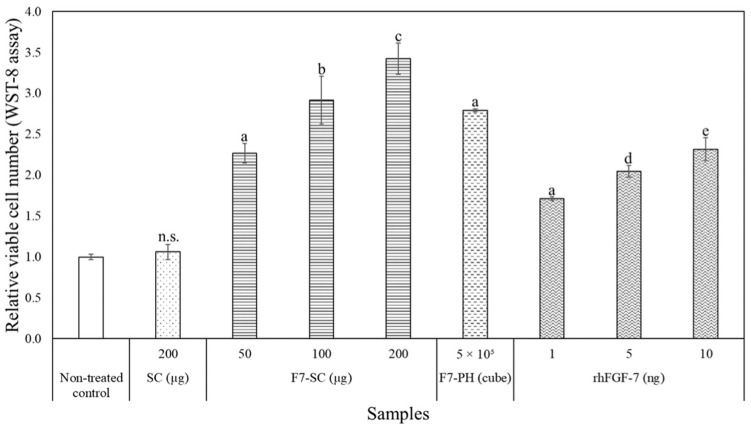
The proliferation of NHEK cells induced by H1/FGF-7. NHEK cells were seeded onto a 24-well plate at a density of 7500 cells per 500 µL of DK-SFM. The culture was then supplemented with SC powder, F7-SC powder, F7-PH crystals, or rhFGF-7 and incubated at 37 °C and 5% CO_2_ for 3 days. The amounts of supplemented samples are indicated on the horizontal axis. Cell proliferation rates were determined as viable cell numbers (as measured using the WST-8 assay) relative to non-treated control culture with PBS only. Data are presented as means ± standard deviations (SDs) of triplicate assays. ^n.s^ *p* > 0.05 vs. non-treated control; ^a^ *p* < 0.001 vs. non-treated control; ^b^ *p* < 0.01 vs. 50 µg of F7-SC; ^c^ *p* < 0.001 vs. 100 µg of F7-SC; ^d^ *p* < 0.001 vs. 1 ng of rhFGF-7; and ^e^ *p* < 0.001 vs. of 5 ng of rhFGF-7.

**Figure 5 ijms-23-09953-f005:**
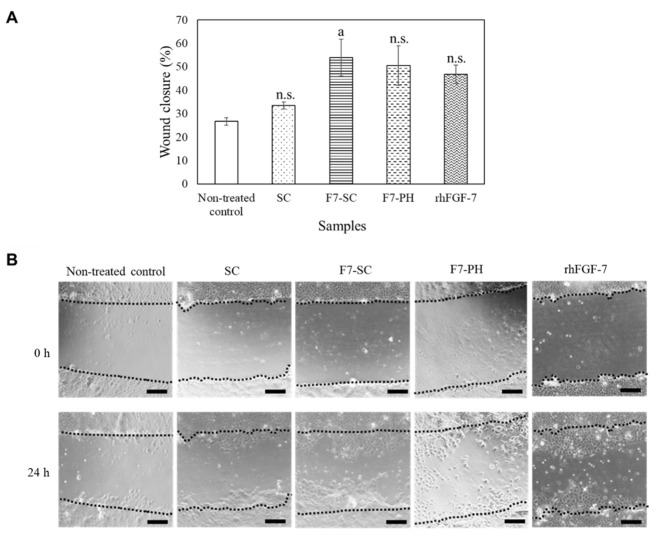
Migration of NHEK cells induced by H1/FGF-7. NHEK cells at a density of 7500 cells per 500 µL of DK-SFM were incubated at 37 °C and 5% CO_2_ for 16 h to induce starvation and then wounded via scratching. The wounded cultures were treated with 100 µg of SC powder, 100 µg of F7-SC powder, 5 × 10^5^ F7-PH, or 10 ng of rhFGF-7, or PBS was added (for non-treated control); then, they were further incubated for 24 h. (**A**) The percentage of wound closure in the scratched NHEK cultures was determined through ImageJ analysis of the photos taken at 0 and at 24 h of treatment. Data are shown as means ± standard deviations (SDs) of triplicate assays. ^n.s^
*p* > 0.05 vs. non-treated control and ^a^
*p* < 0.05 vs. non-treated control. (**B**) Photos of the scratched cultures for ImageJ analysis. Scale bar, 200 μm.

**Figure 6 ijms-23-09953-f006:**
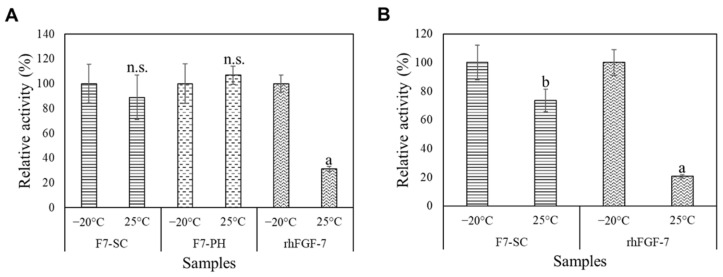
Storage stability of H1/FGF-7 incorporated into sericin-cocoon powder. (**A**) Short-term (7 days) storage stability. Two identical sets of samples including suspensions of F7-SC powder (2 mg/mL), F7-PH crystals (1 × 10^7^ cubes/mL), and a solution of commercial rhFGF-7 (50 µg/mL) were separately stored at −20 °C or 25 °C for 1 week prior to the assays. NHEK cells at a density of 7500 cells per 500 µL of DK-SFM were supplemented with 100 µg of F7-SC powder, 5 × 10^5^ F7-PH crystals, or 10 ng of rhFGF-7. Proliferative activity of the samples was determined as viable cell number measured in the respective cultures after 3 days of cultivation at 37 °C and 5% CO_2_. The proliferative activity of the samples stored at −20 °C was set to 100%, and the relative activity of the corresponding samples stored at 25 °C was determined. (**B**) Long-term (3 months) storage stability of H1/FGF-7. The parameters and analysis method were the same as the 1-week analysis except for sample storage duration. Data are shown as means ± standard deviations (SDs) of triplicate assays. ^n.s^
*p* > 0.05 vs. −20 °C counterpart; ^a^
*p* < 0.001 vs. −20 °C counterpart; and ^b^
*p* < 0.01 vs. −20 °C counterpart.

**Figure 7 ijms-23-09953-f007:**
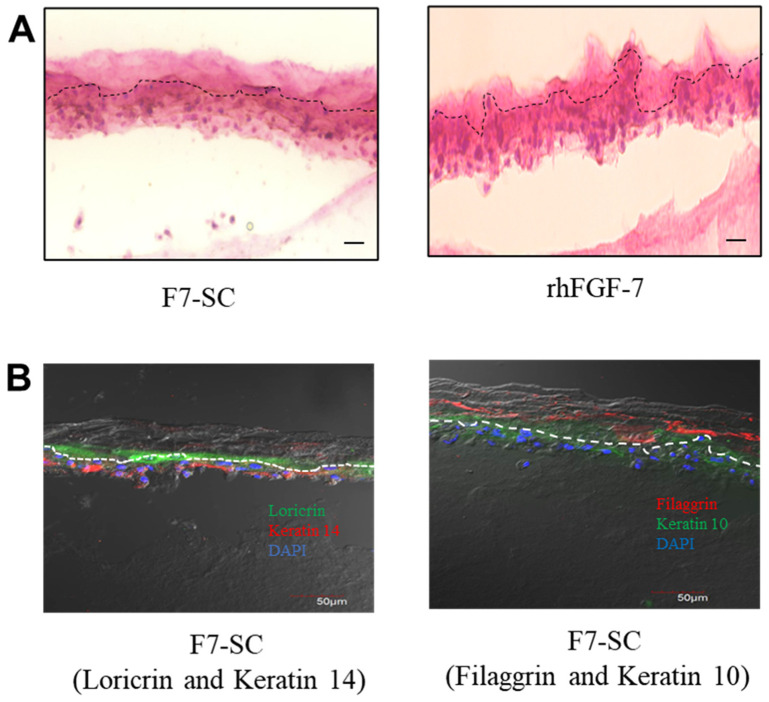
Three-dimensional (3D) cultivation and differentiation of NHEK cells. NHEK cells were cultivated for 2 days on collagen gel under a submerged condition and continued to grow at the air–liquid interface in a medium containing 1.2 mM Ca^2+^ with regular medium change until day 14. The 3D-cultivated cell culture was cryo-sectioned for analysis. (**A**) Hematoxylin and eosin (HE) staining of NHEK cells 3D-cultured (left panel) on collagen gel containing 800 µg of F7-SC powder or (right panel) using basal medium supplemented with 300 ng of rhFGF-7. Scale bar, 20 µm. (**B**) Immunofluorescent staining for expression of differentiation markers on NHEK cells 3D-cultured on collagen gel containing F7-SC powder. Detection of (left panel) loricrin and keratin 14 and (right panel) filaggrin and keratin 10. Nuclei were stained with 4′,6-diamidino-2-phenylindole (DAPI). Upper layer above the dashed line shows the speculated stratum corneum. Scale bar, 50 μm.

## Data Availability

The data presented in this study are available upon request from the authors.

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
