# Peer review of "A Bioengineering Approach for the Development of Fibroblast Growth Factor-7-Functionalized Sericin Biomaterial Applicable for the Cultivation of Keratinocytes"

_ijms, 2022, doi:10.3390/ijms23179953_

Round 1

Reviewer 1 Report (Previous Reviewer 2)

The authors present the revised version of their manuscript. The authors have tried to make an effort to improve the manuscript. However, the manuscript continues to have points of much debate. The results of the cell cultures are arbitrary, the images do not allow us to have such a clear idea of the results.

The reviewer does not understand the supplemental material ? In WB, where are the molecular weight markers? where is the weight of protein displayed?

A very important point is the discussion, in this area of knowledge there are many references. Therefore, the discussion must be independent and with an adequate structure.

This manuscript tries to give very categorical conclusions, but this manuscript is a proof of concept.

Author Response

Response to Reviewer 1.

Thank you very much for the review and comments. The manuscript has been proofread by the English Editing Service recommended by IJMS.

Comments from Reviewer 1:

The authors present the revised version of their manuscript. The authors have tried to make an effort to improve the manuscript. However, the manuscript continues to have points of much debate.

  1. The results of the cell cultures are arbitrary, the images do not allow us to have such a clear idea of the results.

>Thanks for your comments. Data presented for the cell culture experiments (keratinocyte proliferation assays as shown in Figure 4, keratinocyte migration assay as shown in Figure 5) were the mean values of triplicate assays. The HE staining and immunostaining results for the 3D epidermal models (Figure 7) were representative pictures of 3 models generated.  

  1. The reviewer does not understand the supplemental material? In WB, where are the molecular weight markers? where is the weight of protein displayed?

            >Thank you for the comment. In the western blot, we showed the molecular weight marker at first lane on the left of the membrane picture (Lane M). To improve the clarity of the result, we added arrow heads to show the position of the detected proteins. Modification was made to the WB photos in the supplementary materials (Figure S1B) and also the manuscript (Figure 2).

  1. A very important point is the discussion, in this area of knowledge there are many references. Therefore, the discussion must be independent and with an adequate structure. This manuscript tries to give very categorical conclusions, but this manuscript is a proof of concept.

> We thank your suggestion for improvement of the manuscript. We have revised the manuscript by separating the results and discussion. The discussion (Line 312 to 404) is now constructed base on a thought to proof the concept of preparing growth factor-functionalized sericin materials that will be applicable for cell cultivation systems.   

Reviewer 2 Report (Previous Reviewer 3)

Authors have addressed the queries pointed out by reviewers, hence, my recommendation for acceptance.

Author Response

Response to Reviewer 2.

Comments from Review 2:

Authors have addressed the queries pointed out by reviewers, hence, my recommendation for acceptance.

Thank you very much for the review and the comments on our manuscript. In future we will apply the sericin-based biomaterials to the 2D and 3D keratinocyte cultivation systems.

Round 2

Reviewer 1 Report (Previous Reviewer 2)

The authors have partially improved the manuscript. Now yes, the authors have made substantial changes. However, there are still points of major changes:

-In the first place, in the introduction there is still the need for a greater justification of the study. The authors should emphasize this point more.

-Authors should review the WB figure, please show more details. Could the authors show the optical densities? Make comparisons?

-Figure 3 has errors, on day 3 group 2 does not show the bars. Improve the form of presentation, the axes have grammatical errors.

-Figure 4 is still of very poor quality and is not well explained. The reader can see it confused.

-The authors should enlarge panel B of figure 5. It is not well understood.

-In figure 7, the authors should show many more histological images, and with comparative details. Please, this figure should be adequately described in the results section from a more histological perspective.

-The discussion has improved, but it should continue to expand from the perspective of tissue engineering applied to biomedicine. It is still very poor.

-The authors must include an extensive review of the use of English grammar.

Author Response

Response to reviewer.

Comments from Reviewer 1:

The authors have partially improved the manuscript. Now yes, the authors have made substantial changes. However, there are still points of major changes:

>Thanks for the review. We have made the following changes in accordance with your comments.

-In the first place, in the introduction there is still the need for a greater justification of the study. The authors should emphasize this point more.

>Thanks for the comment, we justified the need for the current study by adding

Line 39 to 41

“Long-term availability of biologically active growth factors is required to prolong specific cellular activity which is important for a successful tissue engineering application [9,10].”

-Authors should review the WB figure, please show more details. Could the authors show the optical densities? Make comparisons?

>Thanks for the comment. We have analyzed the band density and presented the band density ratio to

Line 135 to 140

“The band intensity ratio of H1/FGF-7 detected with 5 × 104 F7-PH and 10 µg F7-SC was 1:1.3, as shown by ImageJ analysis. This result implies the amounts of F7-PH and F7-SC used in the Western blot analysis contain comparable level of H1/FGF-7. In order to use approximate amount of H1/FGF-7 from F7-PH and F7-SC for the following biological activity assays, an amount ratio of F7-PH and F7-SC similar to that applied in the current Western blot analysis was used.”

-Figure 3 has errors, on day 3 group 2 does not show the bars. Improve the form of presentation, the axes have grammatical errors.

>Thanks for the comment. Day 3 of group 2 (release profile in DK-SFM) has an error bar, it may not obvious due to the relatively small standard deviation. We have edited the axes of the graph in Figure 3.

-Figure 4 is still of very poor quality and is not well explained. The reader can see it confused.

> Thanks for the comment. To improve the quality of the graph in Figure 4, we have grouped the bars by assigning different patterns to the bars. 

-The authors should enlarge panel B of figure 5. It is not well understood.

> Thanks for the comment. We have enlarged Figure 5.

-In figure 7, the authors should show many more histological images, and with comparative details. Please, this figure should be adequately described in the results section from a more histological perspective.

>Thanks for the comments and the suggestion to improve result’s description. At the present study, a control (cultivation with rhFGF-7) was included in the main text for comparison, and keratinocyte model 3D cultured with F7/poly-SC was presented in Supplementary materials (Figure S1G).

To improve the description for the results, we added dashed lines to the pictures in Figure 7 to separate the speculated stratum corneum layers from other layers with nucleated cells. Moreover, to improve the preciseness of the results description, the sentences:

Line 276 to 278 (in the original manuscript)

“that the culture treated with native rhFGF-7 did not exhibit a clear layer of enucleated cells (Figure 7A, right panel), indicating that the culture did not form a stratum corneum layer containing only enucleated cells.”

was modified to

Line 287 to 289

“that upper layers composed of enucleated cells was relatively thin and with irregular thickness (Figure 7A, right panel), indicating that a poor stratum corneum layer was formed.”

More detailed histological information is not attained with the current stage of study. We added Line 321 to 325 to conclude the results currently achieved with the 3D cultivated keratinocyte and to propose the direction for future work. 

Line 321 to 325

“Results show that the keratinocytes 3D-cultured on collagen gels containing F7-SC were differentiated to form obvious stratum corneum layers. However, further studies to optimize the cell seeding density and media composition would be required for generation of an epidermal model with distinguishable nucleated cell layers which comprise stratum granulosom, stratum spinosum, and stratum basale [36].”

-The discussion has improved, but it should continue to expand from the perspective of tissue engineering applied to biomedicine. It is still very poor.

> Thanks for the comment. We extended the potential application of the growth factor-functionalized sericin materials to tissue engineering by adding

Line 418 to 422

“Additionally, the possibility to prepare allergy-free sericin powders from fibroin-free cocoon shells is beneficial for the in vivo applications. Therefore, the growth factor functionalized sericin materials has a potential to be used as a protein carrier for tissue engineering which require long-term availability of specific growth factors to induce the regrowth of diseased or injured tissues.”

-The authors must include an extensive review of the use of English grammar.

>The whole document has been proofread by using the IJMS associated English Editing Service. We added the certification of this service at the end of the cover letter.

For the certification, please check the ''author-coverletter.

Round 3

Reviewer 1 Report (Previous Reviewer 2)

The authors have correctly made the changes in the manuscript. The manuscript is ready for publication.

This manuscript is a resubmission of an earlier submission. The following is a list of the peer review reports and author responses from that submission.

Round 1

Reviewer 1 Report

I found this work complete and interesting.

I have only few points to address:

1) figure 3: I suggest a legend for the two

curves within the graph.

2) page 5 lines 181-191 I had some problems

to follow the text. I suggest you revise the style 

of the text.

3) I would stress more possible applications 

of the sericin and your results for possible

applications.

4) if you are interested in other examples of 

silkworm silk fibers please have a look also to 

works, more related to mechanical properties

https://doi.org/10.1098/rsfs.2015.0060

DOI: 10.1038/srep18222

Reviewer 2 Report

Lian et al. present a manuscript that may be interesting in conception. However, this manuscript has important limitations. I have read with great interest this manuscript from a promising perspective in tissue engineering, but in reality this manuscript seems to be a very limited proof of concept with conceptual problems.

The authors must implement more important characterization studies, accurately demonstrating the characterization of the possible biomaterial. Second, a small proof of concept of biocompatibility is essential.

This manuscript may be interesting as a general proof of concept in the field of materials, but not from a tissue engineering perspective. The conclusions are categorical and are not supported by purely preliminary results.

The authors must orient the manuscript towards another field, and carry out a discussion of the proposed field of Tissue Engineering.

Reviewer 3 Report

Lian et al. generated a transgenic silkworm line that simultaneously expresses and incorporates FGF-7 into cocoon shells containing almost exclusively sericin. It is claimed that growth factor-functionalized sericin cocoon shells can be extracted by lyophilization and pulverization. The biological activity of the growth factor in stimulating cell growth and migration was characterized. The potential role of the FGF-7 in the cultivation of a 3D epidermal model was demonstrated. Overall, it is a well-designed and presented work. There are some questions related to the biological activity of the growth factor generated by the proposed method. 

1. As shown in the western blot figure, a thick band can be observed from 10 μg sericin cocoons derived from S1sp-F7 × FH-P1A line (lane 4, Figure 2). Could it be possible to determine the percentage of H1/FGF-7 in the sericin cocoons?

2. Figure 3 shows the release profile of H1/FGF-7 from F7-SC powder suspended in aqueous media. It is important to characterize the release profile of the growth in other commonly used cell culture media. 

3. The effective concentration of F7-SC needed in cell proliferation and migration (Fig.4,5) is significantly higher than rhFGF-7. One would question the activity of the growth factor produced from the sericin cocoon. Please explain.